# A Streaming Data Processing Architecture Based on Lookup Tables

**Aximu Yuemaier** [1,2], **Xiaogang Chen** [1,*], **Xingyu Qian** [1], **Weibang Dai** [1], **Shunfen Li** [1] **and Zhitang Song** [1]

1 Shanghai Institute of Micro-System and Information Technology, Chinese Academy of Sciences, Shanghai 200050, China; aximuyme@shanghaitech.edu.cn (A.Y.)
2 School of Physical Science and Technology, ShanghaiTech University, Shanghai 201210, China
* Correspondence: chenxg@mail.sim.ac.cn

**Abstract:** Processing in memory (PIM) is a new computing paradigm that stores the function values of some input modes in a lookup table (LUT) and retrieves their values when similar input modes are encountered (instead of performing online calculations), which is an effective way to save energy. In the era of the Internet of Things, the processing of massive data generated by the front-end requires low-power and real-time processing. This paper investigates an energy-efficient processing architecture based on table lookup in phase-change memory (PCM). This architecture replaces logical-based calculations with LUT lookups to minimize power consumption and operation latency. In order to improve the efficiency of table lookup, the RISC-V instruction set has included extended lookup and data stream transmission instructions. Finally, the system architecture is validated by hardware simulation, and the performance of computing the fast Fourier transform (FFT) application is evaluated. The proposed architecture effectively improves the execution efficiency and reduces the power consumption of data flow operations.

**Keywords:** phase change memory; lookup table; processing in memory; RISC-V ISA; hardware architecture

## 1. Introduction

The Internet of Things (IoT) is increasingly being used in social life and is one of the key industries that is developing in countries around the world. The sensing layer is at the end of the IoT and is a key link in realizing smart networks. Issues such as limited power supply, excessive power consumption, and difficulty in replenishing energy have always restricted the wider application of the IoT. Especially in special environments such as high altitudes, cold temperatures, islands and reefs, it is very difficult to supply power or replace batteries [1]. In addition, a significant amount of power consumption at the front end is wasted on processing invalid signals; for example, in defense and military applications where remote real-time monitoring is often required, more than 90% of the power is lost in sensing and processing useless signals.

Although many studies have proposed zero-power sensors that can filter useful signals at the sensor end and wake up subsequent systems to work, this method only filters signals at the physical level and the processing of data streams still puts a lot of pressure on the front-end controller. At the edge node, it is mainly composed of sensors, microcontrollers, and wireless transceivers, among which the power consumption of transceivers and standby is relatively high [2]. Although microcontrollers have certain data computing capabilities, the calculation of data streams at the front end often involves computationally intensive operators, such as floating-point operations. The power consumption of these computing units is high and requires more clock cycles, resulting in higher latency [3].

In order to solve the problems of energy consumption and latency, memory-centric computing has become one of the effective solutions. Among them, computation based on

lookup tables (LUT) is a way to exchange storage for computation. In LUT-based computation, reading stored values requires less energy than recalculating stored values and reading may also be faster, especially for complex functions. Recently, there have also been many implementations of LUT-based processors [4–7]. At the same time, its reconfigurability provides flexibility in computational capabilities [5,8,9], enabling the implementation of various types of operators in applications, such as linear algebra operations and nonlinear function operations.

Although LUT-based computing has many advantages, the exponential relationship between the capacity of the lookup table and the bit width of the input data makes this computation extremely resource-intensive in terms of storage. However, in low-precision computation, as reflected in the rapidly developing field of approximate computation [10], it is becoming an increasingly important part of modern computational workloads. Research has shown [11] that many applications (such as machine learning, pattern recognition, digital signal processing, robotics, and multimedia) sometimes only require a fairly rough estimate of functions. Other studies have attempted to improve the utilization of lookup table storage space to alleviate the storage capacity pressure [12–14].

At the same time, the development of high-density, high-speed read/write phase change memory [15] (PCM) and other non-volatile memory (NVM) provides an ideal solution for LUT-based computation. The high-speed random access capability of phase change memory can support table lookup operations, and its non-volatility ensures that the lookup table can be saved when power is lost, without having to be read from the off-chip every time power is turned on. More importantly, the high storage density of phase change memory provides a richer and cheaper storage resource for this solution, making it feasible to use table lookup computation in front-end devices.

This paper proposes an LUT-based streaming data processor (LSDP) architecture that uses phase change memory as the on-chip storage medium for lookup tables and instructions and extends the table lookup and streaming data transfer instructions based on a subset of the RISC-V instruction set. Finally, the simulation analyzes the latency and energy consumption of the proposed system for table lookup computation of different types of operators.

## 2. Methodology

### 2.1. Stream Data Processing

Stream processing can be defined as a computer science paradigm that processes a series of data (streams) and a series of operations applied to each element in the stream [16]. In most cases, stream processing is associated with real-time processing. Therefore, stream data processing is a natural solution for Internet of Things (IoT) applications [17]. For example, typical real-time stream processing should include: event detection (collection, filtering, prediction, etc.); real-time data discovery and monitoring; anomaly detection; performance, scalability, and real-time responsiveness. For the front-end streaming data, if it can be processed closer to the data source, it can reduce the power consumption of data transmission and improve the real-time corresponding capability. Most front-end processors process data by reading and writing memory, which is obviously not suitable for processing stream data and can cause unnecessary resource waste and latency. Therefore, after adding the proposed LSDP between the sensor and the controller, most of the processing of stream data can be completed at this layer, as shown in Figure 1, only when complex control logic needs to be processed will the controller be awakened. Through LSDP's data preprocessing or detection capabilities, unnecessary data transmission power consumption can be avoided at the controller end.

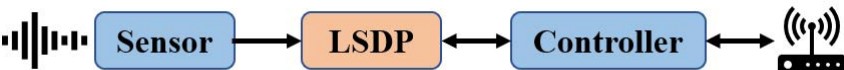

**Figure 1.** LSDP connectivity on the front-end node.

### 2.2. Lookup Table-Based Computing

In LSDP, most of the operations are performed by table lookup. Computing any function requires time and other resources, such as energy. If a specific function value, such as $f(a)$, is needed multiple times in different calculations, storing the calculated value and directly obtaining it from memory when $f(a)$ needs to be calculated, rather than recalculating it. These values can be stored using traditional tables, which are accessed by the values of the operands used as table indices.

An LUT can be viewed as a function $f : I \rightarrow O$; mapping elements from the input set $I$ to the output set $O$. If the number of bits describing an element x ($x \in I$) is represented as $B(I)$, and the bit width is given by $B(I) = \log_2 |I|$. We assume that the bits used to describe the elements of $I$ are in a numerical format so that arithmetic operations can be directly performed on the bits. For example, if I is an integer from 0 to 255, then $B(I) = 8$. If $I$ is an IEEE-754 single-precision floating-point number, then $B(I) = 32$. If it is an 8-bit floating-point number, then $B(I) = 8$. Therefore, the LUT has $B(I)$ bits as input indices and outputs $B(O)$ bit results, and the size of the LUT is $2^{B(I)} \times B(O)$ bits. Figure 2 shows a schematic diagram of a simple direct table lookup method.

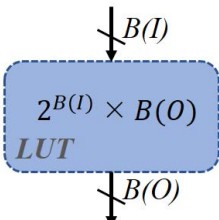

**Figure 2.** Schematic diagram of the direct lookup table method.

Table 1 shows the storage space and LUT size required for the computing results of operators with different data bit widths. When using the direct table lookup method, under general storage resource constraints, when the input bit width is less than or equal to 14, its calculation results can be tabulated. This has advantages in terms of calculation speed and power consumption for floating-point operations or complex function calculations. In addition, thanks to the low cost and higher density of storage devices, as well as improved algorithms that reduce the required LUT size through preprocessing and postprocessing schemes, lookup tables are becoming a viable method for use in higher-precision calculations [18].

**Table 1.** The required LUT storage space for an equation at different input bit widths.

| Bit-Width (Bit) | Memory Bit-Width (Bit) | LUT Size |
| :---: | :---: | :---: |
| 4 | 4 | 32 B |
| 8 | 8 | 512 B |
| 10 | 16 | 2 KB |
| 12 | 16 | 8 KB |
| 14 | 16 | 32 KB |
| 16 | 16 | 130 KB |

## 3. System Hardware Architecture

The proposed LUT-based computational architecture operates as a standalone processor, processing the data stream from the sensor in real-time and outputting the processed data stream or agreed feedback signal. The processor is executed completely sequentially, with no register renaming or instruction-level parallelism. Although in traditional general-purpose processors these LUTs can be stored in main memory and accessed through a data caching mechanism, power consumption, low cache size and hit ratio are major problems for applications where lookups are very frequent. The proposed LSDP hardware architecture is shown in Figure 3.

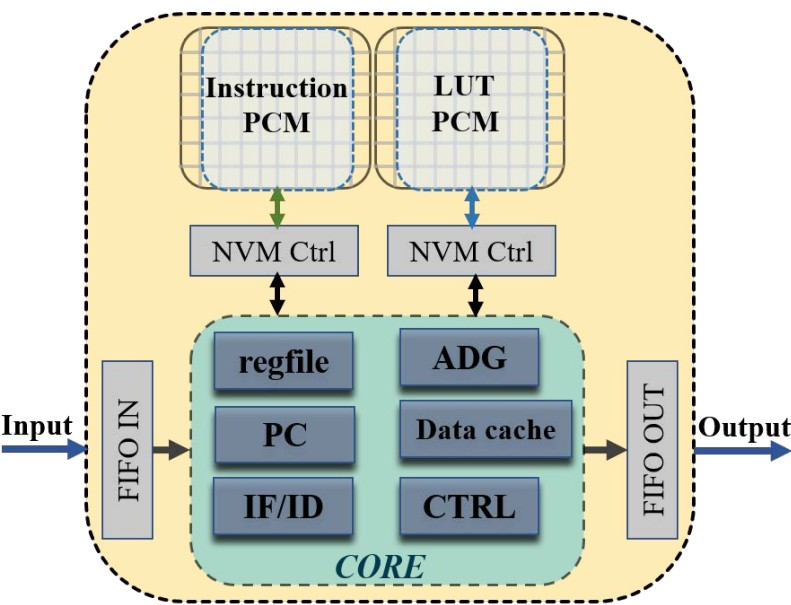

**Figure 3.** Architectural overview of LSDP.

### 3.1. Hardware Architecture Design

The proposed architecture mainly consists of a processor core and instruction and LUT storage units and peripheral devices. The instructions and LUT are stored in two different PCMs, respectively, and can read and write to both memories simultaneously. The memory for storing instructions is composed of several instruction areas in different address spaces, such as interrupt processing instruction area, operation processing instruction area, idle processing instruction area and startup firmware. The memory stored by LUT is composed of LUT base address information table and other LUT space parts. When computing by lookup table, first generate the address of the lookup table through the LUT base address information table and the operator, and then directly read the computing result in the memory through the generated address. Due to the limitation of the PCM interface, the parallel lookup table here is only for the combination of several operators to generate the address when the data bit width is small. As such, the calculation results of multiple operators can be carried out only by looking up the table once. Of course, this also requires the corresponding LUT configuration. The external input and output use a streaming point-to-point bus interface, and the internal instructions can directly take data from the input FIFO or write to the output FIFO for transmission.

The system processor core (CORE) adopts a streamlined three-stage pipeline structure. Regfile is composed of 32 general purpose registers, and PC is the program counter. Data cache is an SRAM module for data storage, with a size of 256 KB. It is divided into instruction fetch (IF), instruction decode (ID), and execution processes. In the instruction fetch process, the fetched instruction is transferred to the corresponding register for further decoding. In the decode stage, further decoding operations are performed on the instruction. In addition, this stage also needs to obtain the corresponding data in advance and put it into the general register for subsequent processing. In the execution stage, unlike general-purpose processors, its execution operation is reading and writing to memory. The specific address required for lookup is generated by the address generator (ADG); for example, when floating-point arithmetic operations need to be calculated, it generates the corresponding address based on the data to obtain the calculation result through reading memory; finally, the processing of data results is also completed in this stage, such as placing it in a data register or directly outputting it to the FIFO.

In order to communicate data during execution, the input and output data buffers of LSDP are associated with a set of different memory-mapped registers. Writing to these registers is buffered using a FIFO, and LSDP can read this FIFO during execution. If the FIFO

is empty, LSDP will stop execution until data are available. Similarly, the output by LSDP is also buffered and can be read by a host or peripheral device connected to the bus. This enables LSDP to support many commonly used streaming media operations in IoT applications.

### 3.2. Processor Instruction Set

#### 3.2.1. RISC-V Instruction Set

For the instruction set of this processor, the RV32I base instruction set of RISC-V is used. The simplicity and regularity of the instructions bring improvements in system power consumption performance [19]. Since the main operation of this system is table lookup and the data transfer method is different from traditional processors, in order to achieve more efficient table lookup computation efficiency, we extended the RISC-V instruction set and added table lookup instructions and data transfer instructions as shown in Table 2. This allows the new instructions to effectively handle table lookup operations on data streams without increasing hardware costs. Figure 4 shows the modified data path of a single-cycle RISC-V processor that implements this instruction set. In addition to the ALU unit and shifter that can implement simple logic instructions, a table lookup storage unit was also added. For example, for an 8-bit wide input, each lookup table has 256 entries, with each entry size up to the 32-bit processor word length. In front-end embedded or edge devices considering cost and power consumption, the number and bit width of lookup tables can be adjusted.

**Table 2.** RISC-V instruction set.

|  | Type | Instruction |
| --- | --- | --- |
| Basic instruction set | Logical operations<br>Shifting<br>Conditional branch<br>Unconditional branch | and,or,xor,andi,ori,andi,ori,xori<br>sll,srl,sra,slli,srli,srai<br>bnq,bne,blt,bge,bltu,bgeu<br>jal,jalr |
| **Extend instruction set** | Data transfer<br>Table lookup | pl.t, plu.t, plr.t, ps.t, psu.t, psr.t<br>ptlu.x, plut.adg.x, plut.adgi.x, plutw.x |

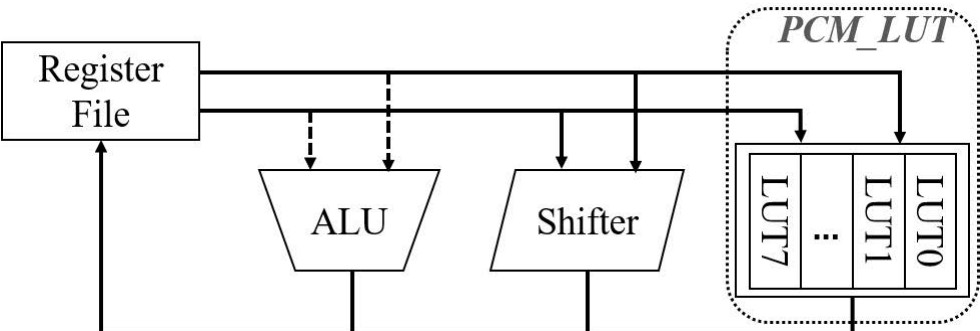

**Figure 4.** Simplified schematic diagram of processor execution with table lookup unit.

#### 3.2.2. Extended Lookup Instruction

The extended table lookup instructions are also divided into three categories: table lookup address generation instructions, table lookup instructions and LUT write instructions. The extended instruction format is mainly based on the R-type, I-type and S-type instruction formats of RV32I, as shown in Figure 5.

| 31 | 25 24 | 20 19 | 15 14 | 12 11 | 7 6 | 0 | |
|---|---|---|---|---|---|---|---|
| funct7 | rs2 | rs1 | funct3 | rd | opcode | | **R** |
| 7 | 5 | 5 | 3 | 5 | 7 | | |

| 31 | | 20 19 | 15 14 | 12 11 | 7 6 | 0 | |
|---|---|---|---|---|---|---|---|
| Imm[11:0] | | rs1 | funct3 | rd | opcode | | **I** |
| 12 | | 5 | 3 | 5 | 7 | | |

| 31 | 25 24 | 20 19 | 15 14 | 12 11 | 7 6 | 0 | |
|---|---|---|---|---|---|---|---|
| Imm[11:5] | rs2 | rs1 | funct3 | Imm[4:0] | opcode | | **S** |
| 7 | 5 | 5 | 3 | 5 | 7 | | |

**Figure 5.** R, I, S type instruction format for RISC-V.

The instruction plut.adg.x for generating a lookup table address is composed of two operand registers to synthesize the address and store it in the destination register:

**plut.adg.x Rd,Rs1,Rs2**

In this R-type instruction, x is a 3-bit encoding in the funct3 field used to distinguish the generation of different bit-width addresses for parallel or block lookup. For example, when both operands are 16 bits, the required LUT size would be too large. In this case, x is set to 4 to divide the 16 bits into four blocks to generate 48-bit addresses. At this time, the four blocks of addresses need to be looked up four times or made into four tables for parallel lookup. If one operand is a constant, use the following I-type instruction:

**plut.adgi.x Rd,Rs1,imm**

The usage of the x encoding is the same as above, and the bit width of the operand and constant must be the same. If there is only one operand, it is equivalent to the immediate number imm being 0, and the instruction plut.adgi.x is still used to handle it.

The format of the lookup instruction is shown below:

**Ptlu.x Rd,Rs1,imm**

The Rs1 operand is the lookup table address generated above, imm is the offset address of the LUT and Rd is the result register of the lookup. The x encoding is also used to determine block lookup or parallel lookup. If multiple tables are stored in a memory with read/write interfaces, the immediate number imm is used as an offset address to read the specified value. The S-type instruction format shown below is used for writing to the LUT.

The write instruction is the same as the RISC-V store instruction, with the only difference being that the x encoding configuration can be used for parallel LUT writing.

### 3.2.3. Extended Data Transfer Instruction

To facilitate the transfer of data streams, the extended data transfer instructions are also divided into three categories: data transfer between registers and input/output and cache. For data transfer between registers and cache, since some lookup operations require intermediate data to be cached and these data only need to stay for one data stream interval, such as FFT operation, therefore, a fast data exchange instruction is required. In the extended data transfer instructions shown in Table 1, pl.t and plu.t are instructions for reading signed and unsigned numbers from the cache to the register, respectively; while plr.t is an instruction for inputting data to a specified register; ps.t and psu.t are instructions for transferring data from the register to the cache, and psr.t is an instruction

for transferring data from the register to the output. The following shows the instruction for reading unsigned numbers from the cache to the register:

**plu.t Rd,Rs1,imm**

The address for data reading is generated by Rs1 and the immediate number imm, and the address is the sum of the register value and the immediate number; Rd is the target write register address. The parameter t is similar to the parameter x in Section 2.2, indicating the division of data. For example, when t is 2, it means that the read 16-bit are is divided into two blocks, that is, there are two results. This division also facilitates the parallel transmission of low-bit-width data, but the bit width that can be transmitted must be a multiple of 4. The other extended instructions interacting with registers are similar to the above and can facilitate the transfer of lookup data streams.

*3.3. FFT Processing Flow Example*

There are many streaming applications in IOT terminals, such as live streaming, video on demand (VOD), audio streaming, podcasting, video conferencing, screen sharing, and gaming. In the processing of speech or video signals, fast Fourier transform is a frequently calculated algorithm. Therefore, we take the FFT algorithm as an example to show the code of the algorithm using the table lookup calculation process of this scheme.

FFT is an effective algorithm for calculating discrete Fourier transform (DFT), and it is also the basic calculation method in the DSP system. The radix-2 Cooley–Tukey algorithm is the classic and simplest FFT algorithm [20]. In this paper, we use the lookup table scheme to replace the traditional FFT calculation method, so that the multiplication and addition of floating-point numbers can obtain the computing results without a complex calculation process.

The pseudocode of the FFT computation is represented by Algorithm 1. The main operation process is the butterfly operation in the for loop, which consists of the multiplication, addition and subtraction of complex numbers. The RICS-V assembly code for this butterfly operation is represented by Algorithm 2, and the assembly code represented by the lookup table extension instruction is represented by Algorithm 3. The assembly code mainly consists of the transfer of data and multiplication and addition and subtraction operations. In the assembly code Algorithm 3 of the extended instruction, pl. 1 is the transfer instruction of data the from cache to register, where the number 1 indicates the data bit width type, and in the experiment, we use an 8-bit-wide floating-point number. The ptlu.x is a lookup table operation instruction, where data x indicates that the data bit width is 8 bits wide, and different values representing the execution of different operator types, such as multiplication, addition and subtraction in butterfly operations. When this instruction runs, the corresponding lookup address is first generated according to the operator type and two operands. The calculation result is then read through a lookup table and transferred to the target register. In traditional processors, floating-point arithmetic units are time-consuming and energy-intensive, and have a high footprint on the chip. Through the extension of the lookup table instruction in the LSDP architecture, the results of these floating-point operations can be obtained by accessing the memory, thereby reducing the energy consumption and increasing the processing speed.

---

**Algorithm 1** Pseudocode of the FFT algorithm.

---

1: **function** FFT(X)(.)
2:     $N \leftarrow length(x)$
3:     **if** $N == 1$ **then**
4:         **return** x
5:     even = fft(x[0::2])
6:     odd = fft(x[1::2])
7:     **for** $k$ in $range(N/2)$ **do**
8:         t = even[k]
9:         even[k] = t + exp($-2j \times$ pi$\times$k/N) $\times$ odd[k]
10:        odd[k] = t $-$ exp($-2j \times$pi$\times$k/N) $\times$ odd[k]
11:    **return** even + odd

---

**Algorithm 2** RISC-V assembly code of the FFT algorithm.

---

**.loop:**

1: slli s1, s2, 3
2: flw f0, s0(s1)                                                                     ▷ t = even[k]
3: addi s2, s2, 1
4: flw f1, s0(s4)                                                                        ▷ odd[k]
5: flw f2, exp_table(s1)                                                 ▷ exp($-2j \times$ pi $\times$ k/n)
6: fmul f1, f1, f2
7: fadd f0, f0, f1                                     ▷ t + exp($-2j \times$ pi $\times$ k/n) $\times$ odd[k]
8: fsub f1, f0, f1                                     ▷ t $-$ exp($-2j \times$ pi $\times$ k/n) $\times$ odd[k]
9: fsw f0, s1(s3)
10: fsw f1, s1(s4)
11: blt s2, s1, .loop

---

**Algorithm 3** Extended instructions assembly code of the FFT algorithm.

---

**.loop:**

1: slli s1, s2, 3
2: pl.1 t0, s0(s1)                                                                    ▷ t = even[k]
3: addi s2, s2, 1
4: pl.1 t1, s0(s4)                                                                       ▷ odd[k]
5: pl.1 t2, exp_table(s1)                                              ▷ exp($-2j \times$ pi $\times$ k/n)
6: ptlu.1 t1, t1, t2
7: ptlu.2 t0, t0, t1                                   ▷ t + exp($-2j \times$ pi $\times$ k/n) $\times$ odd[k]
8: ptlu.3 t1, t0, t1                                   ▷ t $-$ exp($-2j \times$ pi $\times$ k/n) $\times$ odd[k]
9: ps.1 t0, s1(s3)
10: ps.1 t1, s1(s4)
11: blt s2, s1, .loop

---

## 4. Evaluation

The proposed LUT-based processor and extended lookup instructions are synthesized and simulated on Vivado 2020 and implemented on the Xilinx Zynq-7020 platform. The processing architecture kernel part of the simulation is realized by the PL part of FPGA, where the memory is connected through the AXI interface. Figure 6 shows the verification hardware system board of the lookup table memory-based computing processor. The processor core and NVM control part are implemented by FPGA, and two 64 M-capacity PCM chips are used for instruction and LUT storage, respectively. The system clock frequency is 50 MHz.

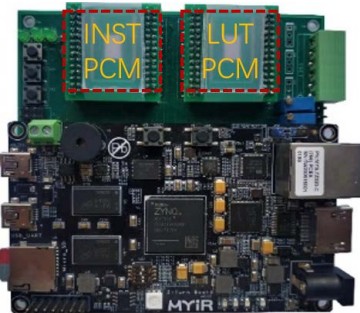

**Figure 6.** Processor hardware emulation environment.

By writing different configuration instructions and reconstructing the LUT, the proposed lookup architecture can support the calculation and mapping of different operation functions, from simple logical functions to complex nonlinear equation operations. Figure 7 shows the average energy consumption and throughput per operand lookup when the data bit width ranges from 4 to bit to 22-bit . We conducted RTL level power consumption analysis on Vivado, where not only the energy consumption of memory access but also the dynamic power consumption of the processor, including the control module, SRAM data cache and registers. In the table lookup computation, the main calculation is obtained through memory access, so we also measured the main electrical parameters of PCM for energy consumption analysis. For the power consumption of PCM, Table 3 shows the electrical parameters when PCM reads data. At a reading voltage of 3.3 V, the power consumption can reach 0.66 nJ/KB. For low-precision operations (such as 4-bit or 8-bit precision), parallel lookup can be implemented using the instruction set to achieve high throughput, and at this time, the power consumption at 4-bit width is only 2.64 pJ/OP. This is because no matter how complex the operation is, there will be a fixed delay when using lookup operations, and the LUT is stored in non-volatile memory, which only needs to be configured once without dynamic power consumption, which makes the energy consumption of each operation very low.

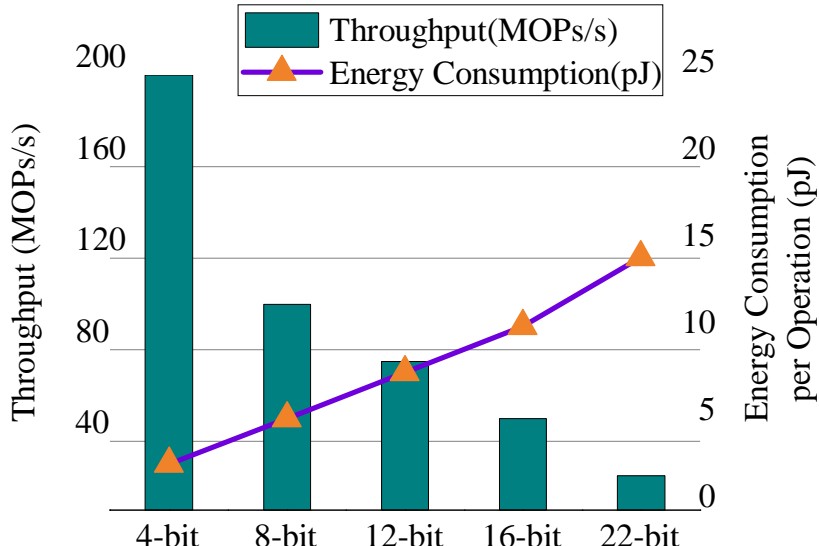

**Figure 7.** Throughput (MOPs/s) and energy consumption (pJ) evaluation of different bit-wide lookup table operations.

**Table 3.** Electrical parameters for PCM data reading.

| $T_{read}$ | $I_{read}$ | $U_{read}$ | $E_{read}$ |
|---|---|---|---|
| 20 ns | 10 μA | 3.3 V | 0.66 nJ/KB |

In order to better demonstrate the performance of the processing architecture, we added experiments to simulate FFT operations in the processor architecture and compared them with the results of FFT processing IP operations in FPGA and other FFT processors, as shown in Table 4. The computation size of FFT is 1024 points. The data bit width is in different forms. In our experiment, the data bit width is an 8-bit floating-point number. Although the data bit width decreases, the accuracy of the FFT will also decrease. However, when using 8-bit floating-point numbers as data, the peak signal-to-noise ratio (PSNR) calculated by FFT can reach 84 dB. The clock frequency of LSDP is set to 50 MHz. Under the same FFT size and clock frequency, the proposed architecture has certain advantages in terms of throughput, computing time, and energy consumption compared to [21], with a power of only 3.53 mW. Due to the high parallelism of computation in FPGA, it has certain advantages in computing speed but consumes more energy than LSDP. In [22], the processor uses a clock frequency of 600 MHz to calculate FFT, and although it has a significant advantage in throughput, it has a relatively high energy consumption. In [23], the data bit width is 16 bits wide, and the clock frequency is only 30 MHz. Although this reduces the energy consumption, it is not very advantageous in terms of computing speed and throughput. In [24], the method of reducing the data bit width was also used. When data are represented as a fixed-point number with a width of 12 bits, there is a high improvement in energy consumption and throughput. However, when represented as a fixed-point number, the computational accuracy is not high, and the processor architecture is relatively difficult to implement in practice. Among other computing schemes with the same frequency, LSDP's throughput has greater advantages. In terms of power consumption, because LSDP is calculated by looking up the table, in addition to the existence of the RISC-V instruction set and lookup table extension instruction set, LSDP has greater power consumption advantages than other calculation schemes.

**Table 4.** Performance comparisons between the proposed architecture and previous designs.

|  | **Our** | **[24]** | **[23]** | **[22]** | **[21]** | **Zynq 7020** |
|---|---|---|---|---|---|---|
| **FFT size (N)** | 1024 | 1024 | 1024 | 1024 | 1024 | 1024 |
| **Word length** | 8/Float | 12-bit | 16-bit | 32-bit | 32/Float | 32/Float |
| **Frequency (MHz)** | 50 | - | 30 | 600 | 50 | 50 |
| **Throughput (MS/s)** | 100 | 890 | 30 | 800 | 0.202 | 6.11 |
| **Execution time (μs)** | 122.88 | - | - | - | 206.44 | 40 |
| **Power (mW)** | 3.53 | 12 | 4.15 | 60.3 | 68 | 44 |

## 5. Conclusions

In this paper, we propose an LUT-based computing architecture in phase change memory for front-end data stream applications. In order to improve the efficiency of data stream transmission and lookup operations, we extended the instructions related to lookup and data transmission based on the RISC-V basic instruction set. Finally, through the analysis of the performance and power consumption of the architecture, it is found that, as the data bit width becomes smaller, the system can achieve higher throughput and reduced delay. Furthermore, while the system has a fixed output delay for calculations, it has a significant advantage in process data streams with small pauses. The research shows that the proposed system architecture has great advantages in processing streaming data with diverse front-end computing forms and strict power consumption restrictions.

**Author Contributions:** Conceptualization, A.Y. and X.C.; methodology, A.Y., X.C. and W.D.; software, A.Y. and X.Q.; validation, A.Y., X.C. and X.Q.; formal analysis, A.Y. and W.D.; investigation, A.Y., X.C. and X.Q.; resources, X.C.; data curation, A.Y.; writing—original draft preparation, A.Y.; writing—review and editing, A.Y. and X.C.; visualization, S.L.; supervision, Z.S.; project administration, X.C.; funding acquisition, Z.S. All authors have read and agreed to the published version of the manuscript.

**Funding:** This project is supported by the Strategic Priority Research Program of the Chinese Academy of Sciences (XDB44010200,XDB44000000-TK-04). It was also supported by Shanghai R&D and Transformation Functional Platform Project (grant agreement number 17DZ2260900): A Functional Platform of Neuromorphic Chips and Intelligent Systems On-chip.

**Data Availability Statement:** The data that support the findings of this study are available from the corresponding author upon reasonable request.

**Conflicts of Interest:** The authors declare no conflict of interest.

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
