# Peer review of "A Streaming Data Processing Architecture Based on Lookup Tables"

_electronics, doi:10.3390/electronics12122725_

Round 1

Reviewer 1 Report

The paper is interesting and timely, but has some weaknesses with regards to evaluation. First, MOPs and latency are very indirect/abstract to have a realistic picture of the performance. Second, the power is not evaluated, which is said to be the main advantage of the approach.
Some concerns relate to big LUTs. An 8GB LUT would have the same latency? (probably not). LUTs of this kind are therefore not useful for many cases, only few (with 8-bit precision, for example). The good thing with the approach is that you can nest LUTs, with instructions, but it is not demonstrated at all, how the instructions impact real-world performance. Figure 7 is problematic. The x-axis is neither linear or logarithmic. Chane it to a bar graph. Other: - PIM is not only for energy efficiency. I also don't see direct correlation with PIM. Maybe this relates more to the memory technology than PIM (change title?)
- "64M-capacity" in bits? - "Dimension" in figure 7 is confusing.  - "fixed delay" how much time? - I am a bit disappointed that the source code is not provided - There are missing details from the evaluation e.g. the MYIR board model, how the memory is connected (are GPIO slow?) Although the evaluation seems to be at early stages, I still see a research contribution. I would like to see a performance comparison for when not using the extended instruction set.

There are some problems with the language use. There is some redundancy (e.g. "in the IoT") and basic mistakes (e.g. ", and islands and reefs,"). For the evaluation, you need to split some text into future work. For example "we can use extended extension instructions": is this future work?

Author Response

In the PDF document.

Reviewer 2 Report

The authors present a compact architecture, based on Processing in Memory (PIM) using phase-change memory (PCM), for low-power and real-time processing of data streams. By employing lookup tables (LUTs) instead of online calculations, power consumption and latency is minimized. Hardware simulation validates the architecture, demonstrating improved execution efficiency and reduced power consumption.

The paper is well written, but could benefit from exploring, for example, succinct data structures that enable efficient traversal of trees, graphs, matrices, and other complex data structures. Notably, the utilization of table lookups for bit manipulations has been well-established, as demonstrated by Jacobson's work on space-efficient static trees and graphs (Jacobson, G., Proc. 30th FOCS, 1989, pp. 549-554). Subsequently, ongoing research has focused on developing dynamic implementations of these data structures, here such a LSDP described in the paper can be helpful to traverse functions that already use the optimum space and optimum time, theoretically, in practical evaluations.

Reviewer 3 Report

The article presents a hardware architecture and an accompanying instruction set extension for RISC-V instruction set aimed at enhancing streaming data processing for IoT devices. The authors design a PIM (Process In Memory) solution that uses PCMs (Phase Change Memory) and LUTs (LookUp Tables) as means to exchange storage for computation.

Section 2.2 must be improved by extending the claim made in lines 104 - 106: “This has advantages in terms of calculation speed and power consumption for floating-point operations or complex function calculations.” The authors are encouraged to extensively present some use cases and measurements that support their claim. Table 1 must be checked (or better explained), because the presented values don’t seem to fit the formula on Figure 2.

On line 128, the authors firstly mention the idea of “compressed instructions” without prior introduction of concept or why did they make such a design choice.

On line 143, the authors vaguely mention: “many commonly used streaming media operations in IoT applications”. The authors are encouraged to clearly state which are these operations and what kinds of algorithms are using them.

In section 3.2, the newly proposed instructions are individually described. Their value would be better emphasized if the instructions were presented in a use-case (with a context). The authors are advised to add some small but relevant (assembly) code snippets or even functions.

The evaluation measures and compares latency and power consumption metrics between various algebraic computations and data sizes. Here, it would be very helpful if some code snippets were presented.

The evaluation must (better) define the metrics that are used: 

  • Latency - between which events?

  • Power consumption - what and how is it measured?

  • Throughput - how is it defined? What do you exactly count/measure?

The latency values in figure 7 are better expressed as: 1 ns, 10 ns, 100 ns, 1us, and so on.

Also, a comparison with the same processing made without extensions (i.e., software only) is recommended. This way, one can clearly see the benefit of using the extensions and the dedicated hardware.

The overall impression is that the authors lightly described their solution without getting into too much detail.

The Quality of English Language is fair.

Reviewer 4 Report

This paper introduces a low-power and high-throughput processor based on PCM-based look-up table. The proposed design is synthesized and implemented on an FPGA board with PCM memory. The authors validated the efficiency of the proposed design by evaluating simple linear algebra operations. Even though the real-hardware implementation sets up a solid foundation for the proposed design, the paper needs to be improved in several aspects as a research paper.

First, the paper lacks some critical details of the proposed design, especially the programmability and processing flow. Specifically, the paper does not illustrate how to program the LUT for different functions and how to utilize the PCM. Without these details, the reviewer found it hard to understand some parts, such as block look-up and parallel look-up. For the parallel look-up, the reviewer wonders if this requires partitioning the PCM for multiple look-up tables. If it is, how do you partition the PCM and how does the chip handle such multi-table lookup? A figure with an example would be greatly helpful.

Second, the architecture configuration of the proposed design is missing, except for the size of PCM. The critical architecture parameters include the PCM bandwidth (or link bit-width), the number/size of registers, the ports of buffers, etc. 

Third, the experiment section needs significantly more details about the implementation. It is critical to see the chip area and energy for all key components in the emulation platform to validate the experimental results. Furthermore, the evaluation method is unclear. How do the authors measure the performance and energy consumption? Simulation or measurement tools on the chip?

Also, the reviewer has several questions about the experimental results. For Figure 7, how does dimension N impact the computation of scalar operations? Also, it is unclear why the latency of matrix addition is constant as N increases. It needs more description of the workloads and how you measure the delay in the experiments.

For Figure 8, it is unclear how do you support 16/32-bit LUT operation. Table 1 shows a 16-bit LUT requires 8GB memory, exceeding 64M of PCM used in your implementation. Again, such confusion results from lack of details about the architecture configuration, processing flow, and experimental setup.

Furthermore, a comparison to some baseline architectures is critical for showing the novelty of the paper. All experiments in the paper only shows absolute value without any comparison. It is hard to justify the usability of the proposed design without knowing if it will outperform existing architectures. The reviewer also suggests some additional experiments to improve the paper. For the energy consumption shown in Figure 8, it would be good to see a breakdown of the different components in the system for each operation. A real application evaluation in the embedded scenario would also strengthen the paper.

Round 2

Reviewer 3 Report

The authors have provided valuable feedback on the formulated requests.

The newly introduced table 4 may require a longer discussion.

Minor editing of English language might be needed.

Reviewer 4 Report

The reviewer appreciates the authors' effort in the revision, which resolves many of the reviewer's initial concerns. However, after seeing the revised version with detailed architecture configuration, the reviewer has some more comments on the architecture design and analysis.

First, the processor core adopts an SRAM-based data cache, with a size of 256MB. This SRAM is very large and can be power-consuming for an IoT-target device. The paper includes neither the justification for such a large SRAM nor the processing flow of using it (e.g., how frequently using the SRAM for a LUT operation).

Second, it seems like the architecture evaluation only considers memory accesses, not including the energy/latency of other components. Even though the latency of other components can be overlapped by the slow memory accesses, the energy consumption should be accumulated into consideration, especially for the large on-chip SRAM and registers.

English looks fine to the reviewer.

Round 3

Reviewer 4 Report

The revision addressed most of the reviewer's concerns.

The English language looks fine. Some minor edits are needed: e.g., the deletion marks in Line141 and Line 260.